# Optimization of Multiple Reactants in a Membrane-Less Direct Methanol Fuel Cell (DMFC)

**DOI:** 10.3390/mi14061247

**Published:** 2023-06-14

**Authors:** Iesti Hajar Hanapi, Siti Kartom Kamarudin, Azran Mohd Zainoodin, Umi Azmah Hasran, Zulfirdaus Zakaria

**Affiliations:** 1Fuel Cell Institute, Universiti Kebangsaan Malaysia, UKM, Bangi 43600, Selangor, Malaysia; iestihajar@gmail.com (I.H.H.); azrans@ukm.edu.my (A.M.Z.); umi.h@ukm.edu.my (U.A.H.); 2Department of Chemical and Process Engineering, Universiti Kebangsaan Malaysia, UKM, Bangi 43600, Selangor, Malaysia; 3School of Materials and Moneral Resources Engineering, Universiti Sains Malaysia, Engineering Campus, Nibong Tebal 14300, Pulau Pinang, Malaysia

**Keywords:** membrane-less direct methanol fuel cell, mixed medium, dual electrolytes, dual oxidants, hydrogen peroxide, multiple reactants

## Abstract

Membrane-less fuel cells are a promising power source for portable applications that enable the solving of membrane-related issues, such as water management and high cost, in conventional fuel cells. Apparently, research on this system uses a single electrolyte. This study focused on enhancing the performance of membrane-less fuel cells by introducing multiple reactants that are dual electrolytes with hydrogen peroxide (H_2_O_2_) and oxygen as oxidants in membrane-less direct methanol fuel cells (DMFC). The conditions tested for the system are (a) acidic, (b) alkaline, (c) dual medium with oxygen as an oxidant, and (d) dual medium and dual oxygen and hydrogen peroxide as an oxidant. Additionally, the effect of fuel utilization on different electrolyte and fuel concentrations was also studied. It was found that the fuel utilization decreases dramatically with the increasing of the fuel concentration, but it improved with the increasing of the electrolyte concentration until 2M. The performance of the dual oxidants in dual-electrolyte membrane-less DMFCs was 15.5 mW cm^−2^ of the power density achieved before optimization. Later, the system was optimized, and the power density increased to 30 mW cm^−2^. Finally, this work presented the stability of the cell using the suggested parameters from the optimization process. This study indicated that the performance of the membrane-less DMFC increased for dual electrolytes with mixed oxygen and hydrogen peroxide as oxidants compared to a single electrolyte.

## 1. Introduction

A fuel cell is a device that generates electricity through oxidation and reduction reactions. It is viewed as having a high capability for use as a green technology due to its unique properties, such as environmental friendliness, high energy efficiency, and long-term energy storage [1]. In addition, Dyer et al. [2] claimed that fuel cell systems have a higher energy density than Li-ion batteries. This technology has had a substantial impact on the research field, as it has been commercialized in applications such as the Toyota Mirai and Toyota FC Bus. Additionally, fuel cell systems have also been implemented in portable devices such as laptops, torches, and portable soldier power [3]. The heart of the fuel cell system is the membrane electrode assembly (MEA) that consists of a membrane sandwiched by electrodes and a catalyst layer. However, membrane-related problems, such as water management, ohmic resistance and electrode poisoning, remain unresolved [4]. Moreover, the presence of a membrane increases the fabrication cost that contributes 20–40% to the overall cost of the system.

The membrane-less fuel cell was first developed in 2002 by Ferigno et al. [5] and solves the membrane-related issues of conventional fuel cell systems. The schematic diagram that portrays the working principle of the membrane-less fuel cell and the conventional PEMFC are presented in Figure 1a,b [6,7]. 

The membrane-less fuel cell adopts a liquid membrane which operates by producing laminar flow that creates a “virtual membrane”, which is the mixing region or inter-diffusion zone. This zone provides ionic conductance to the system to complete fuel cell chemistries [8]. Various types of geometrical designs of this system have been produced. Membrane-less fuel cells are classified into two configurations based on the liquid–liquid interface: side-by-side streaming (e.g., H, T, and Y) and vertically layered streaming (e.g., F and T) [9]. The T-shaped design can exist in both configurations. As described by Bamgbopa et al. [10], vertically layered streaming has a larger contact area between the reactants, thus provide a higher performance. This was proven by Jayashree et al. [11] who found that the F-shaped design generated 26.0 mW cm^−2^ of power density compared to the Y-shaped design with 5 mW cm^−2^. This paper will focus on this configuration, which is the F- and T-shaped design of a membrane-less DMFC.

The special feature of the membrane-less fuel cell is that it can be operated in single and dual-electrolyte media. The dual-electrolyte media can increase the open circuit potential without changing the complexity operation of the system. For a single-electrolyte medium, Whipple et al. [12] utilized a methanol-tolerant catalyst, a ruthenium cluster-like chalcogenide, and generated a peak power density of approximately 4.00 mW cm^−2^. Sun et al. [13] used acidic medium in a membrane-less DMFC to study the effect of the polymer separator. This study proved that this method minimizes the crossover in the system and achieves 7.4 mW cm^−2^. Abrego Martinez et al. [14] generated a power density of 2.16 mW cm^−2^ using nanostructured Mn_2_O_3_/Pt/CNTs as the electrode in acidic medium. Brushette et al. [15] proved the media flexibility of different fuels, and the peak power density of membrane-less DMFCs was generated in alkaline medium (17.2 mW cm^−2^). Thornson et al. [16] used alkaline medium to discover the effect of the electrode length on the performance of membrane-less DMFCs and found that shorter and wider electrodes enhance the performance of a single cell. In another study, Choban et al. [17] studied the media flexibility of a membrane-less DMFC system. A dual-electrolyte medium, also known as a mixed medium, achieves an OCV up to 1.4 V with a power density of 12.0 mW cm^−2^. Chen et al. [18] reported CFD analysis on the performance of membrane-less hydrogen peroxide fuel cell system that operated in dual-electrolyte media and found that the geometric design of the electrode is vital for the performance. In another study, Ponmani et al. [19] developed a membrane-less sodium perborate fuel cell (MLSPBFC) in dual-electrolyte medium, which resulted in a high OCV for the combination of alkaline at the anode and acidic at the cathode side.

Additionally, the oxidant is also an important aspect for the electrochemical reaction in this system. Oxygen is frequently used as the oxidant in the fuel cell system: either pure oxygen that is pumped into the system or atmospheric air (open-air). Jayashree et al. [11] developed a membrane-less direct formic acid fuel cell (DFAFC) that compared open-air breathing with a closed system and found that open-air breathing from the atmosphere generated higher performance. In addition, hydrogen peroxide also has the potential to serve as an oxidant. Hydrogen peroxide is receiving attention among researchers due to its flexibility, as it can have dual roles as a fuel and oxidant. Additionally, it is a simple means of fuel storage, cost-efficient and environmentally safe. However, the behavior of this chemical is not stable in aqueous solution, but it can be stabilized by immersion in acidic solutions, such as sulfuric acid [20]. The reaction kinetics of the oxygen and hydrogen peroxide as oxidants in acidic Equations (1) and (3), and alkaline Equations (2) and (4), electrolytes can be referred to as the following:

Oxygen in acidic medium:(1)$$O_{2}+{4H}^{+}+{4e}^{-}\to {2H}_{2}O$$

Oxygen in alkaline medium:(2)$$O_{2}+{2H}_{2}O+{4e}^{-}\to {4OH}^{-}$$

Hydrogen peroxide in acidic medium:(3)$$H_{2}O_{2}+{2H}^{+}+{2e}^{-}\to {2H}_{2}O$$

Hydrogen peroxide in alkaline medium:(4)$$H_{2}O_{2}+{2e}^{-}\to {2OH}^{-}$$

In 2007, Kjeang et al. [21] used hydrogen peroxide as an oxidant in a membrane-less DFAFC. This study accomplished a power density of 30 mW cm^−2^ and revealed an unsteady stability curve pattern attributed to oxygen evolution from peroxide oxidation. In another study, Rathoure et al. [22] studied the combination of oxygen and hydrogen peroxide in alkaline medium for membrane-less DMFCs. The results showed an improvement in performance compared to the system that operated with oxygen only as the oxidant. Liu et al. [23] utilized hydrogen peroxide as both fuel and oxidant for an electrochemical sensor application which generated 5.5 mW cm^−2^ of power density at 0.66 V.

Although the membrane-less fuel cell was introduced in 2002, a lack of studies in this field persists, especially in membrane-less DMFC systems. Membrane-less DMFCs suffer from low performance due to low oxygen concentrations in the cathode, as low oxidants can disturb the electrochemical reaction. These issues can be reduced by using the uniqueness of the system, namely, media flexibility. The use of hydrogen peroxide as an oxidant is favorable to increase the theoretical open circuit voltage (OCV). It is also beneficial to increase the oxygen concentration in membrane-less fuel cells that suffer from low oxygen concentrations at the cathode side [24].

Apparently, no other researchers have reported the performance of membrane-less DMFCs in dual electrolytes using both H_2_O_2_ and oxygen as oxidants. Therefore, this paper reports on the performance enhancement utilizing the combination of hydrogen peroxide and oxygen (open-air) as the oxidants in a dual-electrolyte membrane-less DMFC with the optimization using response surface methodology (RSM). This work also highlights the effect of the electrolyte and oxidant concentrations on the performance. Moreover, fuel utilization is also a focus of this study, as it is the major issue in this system. The optimization of the system is also described in this paper.

The optimization process response surface methodology (RSM) and the Taguchi method are commonly used by researchers. The RSM is crucial to enhance system performance and optimize process yield while maintaining cost efficiency. The Taguchi approach and analysis of variance methodology have been used to analyze a design parameter and two operational parameters with regard to the performance enhancement of the PEM fuel cell with a 25 cm^2^ active area of interdigitated flow channel [25]. The advantages of the RSM method over the Taguchi method is it is more efficient in predicting the response by using the mathematical modeling [26]. RSM refers to a collection of mathematical and statistical techniques that are applicable for enhancing, constructing, and refining processes in which a targeted outcome is influenced by multiple parameters [27]. RSM is a more effective approach for optimizing process parameters due to its ability to account for the interactive effects among the variables being tested. There are plenty of techniques that can be utilized in the optimization process. RSM with the Box–Behnken design was used by Carton et al. [28] to optimize the parameters affecting the flow field of the PEMFC and found the relation between the voltage and current density that are at maximum 0.97 V and 2.9 A of current. Roudbari et al. [29] utilized the optimization RSM using central composite design (CCD) in evaluating the impact of the parameters on the PEMFC oxygen reduction reaction (ORR) with the maximum power performance 29.63 mW cm^−2^. In the membrane-less fuel cell system, Muaz et al. [30] optimized the membrane-less microbial fuel cell system by using RSM via the CCD method for optimum electrical generation of the system. Recently, Oh et al. [31] optimized the double-bridge flow channel of the membrane-less microfluidic fuel cell and enhanced 57.6% of the power density compared to the reference design. Moreover, in RSM the CCD is superior to the Box–Behnken Design for predicting responses that are closer to the actual values. Therefore, this work utilized the RSM with the CCD method for optimization of the proposed membrane-less DMFC system.

## 2. Methods

### 2.1. Materials

The catalysts used for the cathode and anode were Pt black (HiSPEC 1000, Alfa Aesar, Ward Hill, MA, USA) and Pt-Ru Black (HiSPEC 6000, Alfa Aesar, Ward Hill, MA, USA), respectively. The gas diffusion layer (GDL) was produced using Toray^®^ carbon paper or cloth. The Nafion^®^ and polytetrafluoroethylene (PTFE, TGH-H-060) solution served as a binder to create the hydrophobic electrode. Stainless steel mesh and methanol (MeOH) were used as the current collector and fuel, respectively.

### 2.2. Fabrication

Three types of single cells were fabricated in this study: an F-shaped design with closed-air and open-air systems and a T-shaped geometrical design. The channel, anode and cathode endplates were prepared using polymethylmethacrylate (PMMA). The components were fabricated using an Engraver Machine (Roland EGX-350). The channel dimensions were 30 mm long, 3 mm wide and 2 mm thick. PMMA (8 mm thickness) was used as the anode and cathode endplates. At the cathode, a 15 mm-long and 3 mm-wide window was machined to allow the gas diffusion electrode (GDE) to be exposed to the surrounding air. The difference between the open-air and closed-air F-shaped designs is the presence of a window at the cathode endplate. Silicone was used as the gasket to prevent leakage, while stainless steel was used as the current collector. An electrode with an active area of 0.45 cm^2^ was clamped between the endplates.

### 2.3. Electrode Preparation

The diffusion layer was created using Toray^®^ carbon paper/cloth treated with polytetrafluoroethylene (PTFE, TGH-H-060). The microporous layer (MPL) was prepared by directly painting carbon black (Vulcan black Ec 300J, Lion Corp., Tokyo, Japan) on the carbon paper or cloth. The different backing layers that are carbon paper and carbon cloth are used with the similar method of MPL and catalyst layer (CL) preparation for the performance analysis of the different electrode in Section 3.2. For the CL, each of the electrodes used 8 mg cm^−2^ catalyst loading and the method referred to the previous report [32,33]. Pt and Pt–Ru black were immersed in a 5 wt% Nafion^®^ solution (Wako Pure Chemical Industries, Ltd., Osaka, Japan), isopropyl alcohol and deionized water. Then, the solution was ultrasonically mixed to produce a homogenous mixture and cast on the MPL. The electrode was dried in an oven for an appropriate period.

### 2.4. Fuel Cell Testing

All elements in the single cell were properly aligned in a vertical layer, and distilled water flowed onto the system to test for leaks. If leakage of the system was detected, which is caused by the misalignment of the layers, the system was disassembled and realigned. The anolyte and catholyte were fed into the system using a syringe pump (SPLab01, Shenchen, Baoding, China) at a flow rate of 0.3 mL/min. This membrane-less DMFC system setup for the performance testing by using the potentiostat/galvanostat (WonATech, Seoul, Korea) is presented in Figure 2.

The Reynolds number was calculated to determine the laminar flow of the system. Afterwards, an aqueous solution of iron(II) chloride (FeCl_2_) (Sigma Aldrich, St. Louis, MO, USA) and bathophenanthroline sulfonate (BPS) (Sigma Aldrich, St. Louis, MO, USA) was used to prove that no mixing occurred between the reactants that could reduce the performance. Both chemicals were fed into the system as anolytes and catholytes at a flow rate of 0.3 mL/min using a syringe pump. The change in the color of the solution was observed during the operation.

Next, the single cell with the closed-air F-shaped design was used to analyze the performance achieved by the system operated using different types of electrodes: carbon paper and cloth. Then, a single cell was used to study the effect of the open-air and closed-air systems on performance. Then, the performance of the F- and T-shaped geometrical designs of the membrane-less DMFC was compared under similar operating conditions. A schematic diagram of the membrane-less DMFC system can be referred to in Figure 3.

The single cell that achieved the best performance was used to analyze the effect of the different medium conditions. Four types of operating conditions were studied: acidic, alkaline and dual electrolytes with oxygen as an oxidant. The fourth condition combined oxygen and hydrogen peroxide as oxidants in dual-electrolyte medium. The complete systems for these four conditions are simplified in Table 1. The polarization curve was obtained using a potentiostat/galvanostat (WonATech, Seoul, Korea). The best condition of the system was used to optimize the operating conditions.

The main factors that affect the performance of membrane-less DMFCs were discovered using a one-factor-at-one-time (OFAT) method to approximate the optimization level. The H_2_O_2_, H_2_SO_4_, KOH and MeOH concentrations used by other researchers [22,34,35,36] were chosen to determine the screening level. The concentrations of all chemicals ranged from 0.5, 1.0, 2.0 and 3.0 M for the OFAT process, and the results are presented in Table 2. The OFAT method was utilized for screening to determine the significant effect parameters to the response. The first parameter is the H_2_O_2_ concentration that varies in the range of 0.5 to 2.0 M, whereas the other parameters were constant at 1 M. Then, the best concentration of H_2_O_2_ that generated the highest response was preferred to determine the H_2_SO_4_ parameter (varies from 0.5 to 2.0 M), and the other parameters were at a constant (1 M). This technique was repeated until all of the parameters were evaluated. The central composite design (CCD) within the RSM was utilized to visualize the optimal power generated with the most significant variables: H_2_O_2_, H_2_SO_4_ and KOH concentrations. According to the CCD, 20 runs of experiments were conducted, and the observations were adapted to the second-order polynomial model. The regression coefficients, ANOVA, F-value and *p*-values were analyzed to evaluate the developed model. The statistical software package Design-Expert^®^ 11.1.2.0 (Stat-Ease Inc., Minneapolis, MN, USA) was used to generate the model for predicting the effect of the factors on the response. The suggested parameters were used for the validation process. Last, stability tests were performed using the optimum parameters suggested by the CCD at a 0.6 V potential.

## 3. Results

### 3.1. Determination of Laminar Flow

Commonly, the laminar flow condition is determined by calculating the Reynolds number using Equation (5):(5)$$Re=\frac{\rho \nu L}{\mu }.$$
where ρ is the reactant density, ν is the reactant velocity, L is the channel length and μ is the dynamic viscosity of the reactants. The Reynolds number of this system is approximately 24. This value is under the range of the laminar flow condition. In addition, an observation study was performed using BPS and FeCl_2_ that were fed into the system with a syringe pump at a flow rate of 0.3 mL/min. Both solutions were colorless, but a cherry-red color was produced when they were mixed as presented in Figure 4. The observation analysis also showed a lack of obstacles or burrs that disturbed the reactant flows and proved that no mixing occurred between the catholyte and anolyte along the channel.

### 3.2. Performance of Membrane-Less DMFCs Operated with Different Types of Electrodes

The best operating conditions for the membrane-less DMFC system were determined using an F-shaped design. First, the effect of different types of electrodes was studied using carbon paper and cloth in the system. As shown in Figure 5, the open circuit voltage (OCV) and power density generated by the system utilizing the carbon cloth and paper were 0.19 and 0.23 V with 0.043 and 0.103 mW cm^−2^, respectively. The difference between the single-cell performances is due to the morphological structure. Based on the study by Radhakrishna et al. [37], the morphology of carbon cloth is a woven structure, whereas the carbon paper structure is similar to carbon fibers held in resin. The woven structure of the carbon cloth caused it to become more porous than the carbon paper. The membrane-less DMFC system is operated with the aid of an external pump; thus, the large pore size of the carbon cloth may facilitate reactant crossover and degrade the performance. The best performance was observed for the carbon paper due to the hydrophobic MPL layer, which had a high resistance to water due to the small size of the pore structure. Water accumulation at the electrode surface can be avoided; thus, a larger active area is exposed for the chemical reaction in the system [38].

### 3.3. Performance of Membrane-Less DMFCs Operated in Different Modes

The system utilizing carbon paper as an electrode was assessed to determine the best operating mode for the system. Two modes were investigated: closed- and open-air systems. The open-air system has a window at the cathode endplate to allow the movement of the air from the surroundings. Meanwhile, for the closed-air system, oxygen was bubbled to the catholyte. As shown in Figure 6, the open-air system achieved an OCV of 0.43 V with a power density of 1.689 mW cm^−2^. For the closed-air system, the power generated was 0.103 mW cm^−2^ at an OCV of 0.23 V.

### 3.4. Performance of Membrane-Less DMFCs with Different Geometrical Designs

This study was continued by comparing the performance of the membrane-less DMFC F- and T-shaped designs. These geometrical designs were preferred in this study because they are grouped in a similar configuration of vertically layered streaming. Moreover, this configuration leads to a higher contact area between the reactants and is free from gravity effects on the system [39]. The result presented in Figure 7 shows a higher performance of the membrane-less DMFC with the T-shaped design than the F-shaped design, which is an OCV of 0.53 V with a power density of 2.89 mW cm^−2^ and an OCV of 0.42 V with a power density of 1.68 mW cm^−2^, respectively.

### 3.5. Performance of the Membrane-Less DMFCs in Different Oxidant and Media

Figure 8a,b presents the polarization and power density curves of different media of air-breathing membrane-less DMFCs. The OCVs of the membrane-less DMFCs in acidic and alkaline media were 0.5 and 0.7 V, respectively. No significant differences were observed between these two media. In addition, by referring to the graph, the power densities generated for the acidic and alkaline media were 2.9 and 3.2 mW cm^−2^, respectively. Notably, the system operating in alkaline media exhibited better performance than in acidic media. This difference is due to the increase in methanol oxidation that has been claimed to produce a higher current density than the acidic medium in a previous study. Additionally, the dual electrolytes (mixed-medium) with open air as the oxidant increased the performance by 50% compared to all acidic and alkaline media.

### 3.6. Fuel Utilization

The main problem in this system is low fuel utilization per single pass. This low fuel utilization is one of the factors that causes low power density generation. Fuel utilization, η, was calculated using Equation (6).
(6)$$\eta \;=\;\frac{I}{nFCQ}$$
where *I* is the current at the flow rate *Q*, *n* is the number of electrons released per mole, *F* is Faraday’s constant, and *C* is the concentration of fuel. Using Equation (6), the fuel utilization based on the different electrolytes and fuel concentrations for the best performance of membrane-less DMFCs was plotted in Figure 9.

This study focused on fuel utilization of the system operating with different concentrations of methanol and KOH electrolytes at the anode side. Electrolytes function as bridges in the system to allow the transportation of protons to the cathode side. As shown in Figure 9, fuel utilization increased from 0.39 to 0.85% at electrolyte concentrations of 0.5 to 2 M and decrease when the concentration exceeds 2 M.

### 3.7. Optimization of the Membrane-Less DMFCs

#### 3.7.1. Preliminary Study

The effects of fuel, electrolytes and oxidant concentrations were discovered using the OFAT method. The maximum power density generated for each parameter is recorded in Table 2.

Increasing the H_2_O_2_ concentration from 0.5 to 2 M improved the performance of the membrane-less DMFC from 12.7 to 18.9 mW cm^−2^. The electrolyte acts as a bridge for the movement of ions in the system. Increasing the H_2_SO_4_ concentration from 0.5 to 2 M increased the power generated by the system by 25.9%, but the power density decreased when the concentration increased to 3 M. This effect is due to the increased concentration of the hydronium ion that increases the conductivity for an efficient reaction process and reduces the resistance. Regarding the KOH concentration, the efficient electrochemical reaction in the presence of 2 M KOH generated a power density of 25.4 mW cm^−2^. The power density decreased to 23.9 mW cm^−2^ when utilizing 3 M KOH.

The power density increased from 24.8 to 27.1 mW/cm^2^ as the MeOH concentration increased from 0.5 to 2 M. Thus, mass transfer is the limiting factor at low MeOH concentrations. At an MeOH concentration of 3 M, the performance decreased due to a high level of fuel crossover. All of the parameters affected the power density generated by the system. Among them, the MeOH concentration generated less of an effect; hence, the parameters preferred for the optimization process were H_2_O_2_, H_2_SO_4_ and KOH concentrations.

#### 3.7.2. Central Composite Design

In the preliminary study, the effects of H_2_O_2_, H_2_SO_4_, KOH and MeOH concentrations were studied. H_2_O_2_, H_2_SO_4_ and KOH were labelled A, B and C, respectively, whereas the MeOH concentration was held constant at 2 M. By referring to the result obtained from OFAT, the correlation between variables A, B and C exerted the greatest effect on the power density generated by the membrane-less DMFC. It was continued by using the statistical tool response surface methodology (RSM) via a central composite design (CCD) to determine an interactive effect between the tested variables.

#### 3.7.3. Regression Model Equation and Statistical Analysis

From the OFAT, A, B and C were chosen in Table 3 as factor levels. Table 4 presents the experimental data from Design Expert. By referring to the sequential model sums of squares, the quadratic model was an excellent polynomial model to portray the effects of the independent variables on the response. All responses were analyzed using analysis of variance (ANOVA) and are shown in Table 5.

Parameters A, B, C, AB, AC, A^2^, B^2^ and C^2^ were significant for the power density generated by membrane-less DMFCs. The relationship between the three independent factors fit the quadratic model. Equation (7) shows the analyses of the regression coefficients of the three factors:Y = +29.56 − 0.5076 A − 0.7192 B − 0.5059 C − 0.8250 AB − 0.9500 AC + 0.0750 BC − 1.68 A^2^ − 1.68 B^2^ − 2.16 C^2^(7)

Using the regression model, the predicted data were obtained. The result was compared to the experimental data. Figure 10 shows no significant differences between the predicted values and the experimental data, thus supporting the quadratic model. The normal percentage probability plot of the residual in Figure 11 indicated that the error was normally distributed as the plot fell on a straight line. The plots in Figure 10 and Figure 11 are acceptable, and the empirical model was suitable to interpret power density generation.

The interaction between the predicted model obtained using Equation (7) on the power density generation and the ANOVA shown in Table 5 can also be represented in the response surface plot and contour plot. Figure 12 and Figure 13 present the interaction between the H_2_O_2_, KOH and H_2_SO_4_ concentrations that form a parabolic cylindrical shape. This pattern shows all the interactions presented a peak point in the experimental domain.

#### 3.7.4. The Optimum Range of Process Parameters and Validation of Models

The optimum condition suggested by the software generated a power density of 29.13 mW cm^2^ with a desirability of 0.917. The response and the three independent variables were set at maximum values and in the range of the investigated values. Table 6 shows the validation results of this experimental work.

The means and standard deviations from the experimental runs were determined. The 95% confidence interval was also calculated to compare the predicted value to the experimental value. The means and standard deviations were 29.72 and 0.698 mW cm^−2^, respectively. The 95% confidence interval was 29.092 to 30.356 mW cm^−2^.

Table 7 summarizes the performance of membrane-less DMFCs from the previous literature, including the system investigated in this study.

### 3.8. Stability

Figure 14 shows the behavior of the current density of this system at the optimum parameters as a function of time. As observed in the stability result, the current density decreased exponentially and remained constant. Kjeang et al. [21] claimed that hydrogen peroxide produces an unsteady pattern of stability due to oxygen evolution from the oxidation of peroxide.

## 4. Discussion

As shown in Figure 4, no cherry-red coloration was along the channel of the system, which proved that the solution was not mixed and flowed with a laminar behavior. However, the cherry-red color formed at the outlet for the waste solution of the system due to mixing. This coloration is due to the high affinity of Fe^2+^, which produces a trichelate of BPS [Fe (BPS)_3_]^4−^. Additionally, the difference between the single-cell performances in Figure 5 is due to the morphological structure. Based on the study by Radhakrishna et al. [36], the morphology of carbon cloth is a woven structure, whereas the carbon paper structure is similar to carbon fibers held in resin. The woven structure of the carbon cloth caused it to become more porous than the carbon paper. The membrane-less DMFC system is operated with the aid of an external pump; thus, the large pore size of the carbon cloth may facilitate reactant crossover and degrade the performance. The best performance was observed for the carbon paper due to the hydrophobic MPL layer, which had a high resistance to water due to the small size of the pore structure. Water accumulation at the electrode surface can be avoided; thus, a larger active area is exposed for the chemical reaction in the system [38].

Additionally, the result shows a substantial difference between these two operating modes in Figure 6 because of the shortage of oxygen in the closed-air system. The solubility of oxygen is low in 0.5 M of sulfuric acid: namely, 1.14 Mm [39]. This low solubility caused a low concentration of oxygen as an oxidant in the cathode that disturbed the electrochemical reaction. All of these operating conditions were used in determining the effect of the geometrical design on the fuel cell performance. Figure 7 shows the F-shaped design of the membrane-less DMFC exhibited lower performance than the T-shaped design due to the different distances between the electrodes. The F-shaped design has a special plate as a boundary to prevent the mixing of the reactants. Thus, it increases ohmic resistance by increasing the distance travelled by the ion to reach the active site [11]. The electrode distance affects the resistance and can be reduced when the electrode distance decreases.

The membrane-less DMFCs were operating in different reactants (Table 1). The performance for dual-electrolyte is superior compared to the single-electrolyte operation. This situation was related to the OCV, which was improved in the system using the dual-electrolyte operation. Typical OCV was reported to range from 0.4 to 0.8 V, while in this study, the OCV reached approximately 1.2 V for the dual-electrolyte system using a combination of oxidants (H_2_O_2_ and oxygen). The reaction of mixed-oxidant oxygen and hydrogen peroxide in dual-electrolyte medium can be seen as Equations (8)–(11) that shows a high theoretical voltage for the system. Moreover, the utilization of H_2_O_2_ boosts the performance of the system. The presence of the H_2_O_2_ in the catholyte enhances the chemical reaction in the mixed media used in this system by the presence of additional oxygen molecules for the reduction reaction. The power density reported in this study was 15.5 mW cm^−2^, which is higher than that of the dual-electrolyte system using oxygen as the oxidant. The results presented here proved a serious problem faced by the membrane-less DMFC system, which lacks oxygen in the cathode and triggers decreased performance. Therefore, H_2_O_2_ functions as a good oxidant that acts as an additional oxygen ion for better mass transfer. This is owing to the hydrogen peroxide characteristic that it is very unstable due to its chemical structure and unpaired electrons [40]; thus, it is a very strong oxidizing agent. The oxygen–oxygen single bond (peroxide bond) causes the structure to be unstable and easily decomposed [41].

Reaction of a membrane-less DMFC operated in dual-oxidant hydrogen peroxide and oxygen in dual-electrolyte media:

Anode:(8)$${CH}_{3}OH+{6OH}^{-}\to {CO}_{2}+{5H}_{2}O+{6e}^{-}$$

Cathode:(9)$$O_{2}+{4H}^{+}+{4e}^{-}\to {2H}_{2}O$$
(10)$$H_{2}O_{2}+{2H}^{+}+{2e}^{-}\to {2H}_{2}O$$

Overall reaction:(11)$${CH}_{3}OH+H_{2}O_{2}+O_{2}+{6H}^{+}+{6OH}^{-}\to {CO}_{2}+H_{2}O$$

Moreover, the increase in the OH concentration that moved from the anode to the cathode improved the electrochemical reaction in the system by reducing the ionic resistance [42]. As can be seen in Figure 9, at a 3 M concentration, the fuel utilization decreased to 0.28% due to the excess OH ions that covered the active site and hindered the reaction. However, fuel utilization decreased continuously when the fuel concentration increased. Fuel utilization decreased from 0.94 to 0.12% when the methanol concentration ranged from 0.5 to 3 M. The issue of this system is that the reactants did not fully react, causing low fuel utilization. The electrochemical reaction occurs at the interface of the electrode and electrolyte. Thus, most of the methanol that flows at the center of the stream does not participate in the reaction. The fuel utilization is low, but it can be improved by using lower flow rates. Lowering the flow rate causes a decrease in power density but it can be enhanced by using a high concentration of supporting electrolytes, a higher surface area of catalyst and by using different fuel with greater electrochemical activity [43].

The optimization using RSM via the CCD method was utilized in this study. H_2_O_2_ functions as an additional oxygen ion in the system by adding to the oxidant concentration at the cathode for a better electrochemical reaction. However, the performance decreases at an H_2_O_2_ concentration greater than 2 M due to an excessive amount of oxidant that disturbs the active site of the system [22]. On the other hand, the KOH and H_2_SO_4_ electrolytes also have a similar pattern of a decrease in performance when the concentration exceeds 2 M. The reaction was reduced when an excessive amount of the ion was present at the active site that hindered the reaction of the system. In addition, the high concentration of electrolyte reduced the water concentration in the solution and contributed to the slow chemical reaction. The high concentration of unreacted ions covering the active site prevents hydrogen peroxide and oxygen from reaching the area for the reaction [44]. Additionally, an increase in the electrolyte concentration increases the viscosity of reactants, which may hamper the transportation process, thus increasing the internal resistance [45].

The response surface and contour plots for the effects of H_2_O_2_ and H_2_SO_4_ concentrations on power density are presented in Figure 12a,b. The pattern was consistent with the OFAT result that the generated power density increased with an increasing H_2_O_2_ concentration. This finding might be related to the additional oxygen ions present at the cathode for an efficient reaction. Moreover, the increase in the H_2_SO_4_ concentration also increased the power density as the proton conductivity became more efficient. The other role of H_2_SO_4_ was to stabilize H_2_O_2_ in the system. Similar to the response surface and contour plots of KOH and H_2_O_2_ in Figure 13a,b, the power density generated by the system increased when both of the concentrations increased and decreased at certain points. This result is due to the increase in the conductivity of ions and caused a more efficient electrochemical reaction in the system.

The predicted data were in the range of the tested values. We concluded that the predicted values within the 95% confidence interval and the optimum condition proposed by the RSM were valid. Thus, the RSM performed with the CCD was useful to optimize the operating conditions of membrane-less DMFCs. The performance of the single-cell optimization using RSM via CCD was increased by approximately 10% compared to optimization using OFAT.

Furthermore, the graph in Figure 14 did not fluctuate, as reported in a previous study. This condition represented the optimum condition and successfully stabilized the H_2_O_2_ in the system. According to An et al. [20], the addition of acid (H_2_SO_4_ in this study) not only stabilized the H_2_O_2_, but also facilitated the electroreduction process. This result proves that the optimization process was accepted as it able to stabilize the working principle of this membrane-less DMFC. The system could supply a constant current density of 10 mA cm^−2^ over time.

## 5. Conclusions

Membrane-less DMFCs have high potential for use in portable power applications. This study focused on the vertically layered configuration of geometrical designs that are F- and T-shaped. However, the low performance of the membrane-less fuel cell system halts the commercialization of the application. Therefore, this study highlighted the performance enhancements for the membrane-less DMFCs by discovering the reactants used for the anolyte and catholyte. Typically, fuel cell research employs a single-electrolyte medium. However, this particular investigation utilized a dual-electrolyte medium and incorporated a dual oxidant (O_2_ + H_2_O_2_). The findings indicate that this approach has the potential to produce optimal performance. Additionally, the reactants used as the anolyte and catholyte were optimized using RSM, and the optimum power density for the system was generated at 30 mW cm^−2^ at H_2_O_2_, H_2_SO_4_ and KOH concentrations of 1.90, 2.1 and 1.2 M, respectively.

## Figures and Tables

**Figure 1 micromachines-14-01247-f001:**
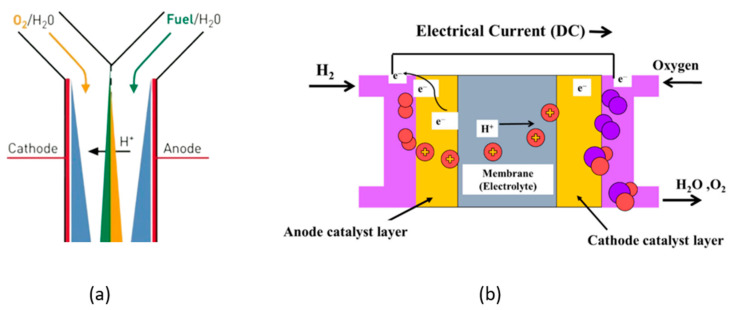
Schematic diagram of (**a**) membrane-less fuel cell Reprinted/adapted with permission from [6]. 2003 Elsevier and (**b**) conventional PEMFC [7].

**Figure 2 micromachines-14-01247-f002:**
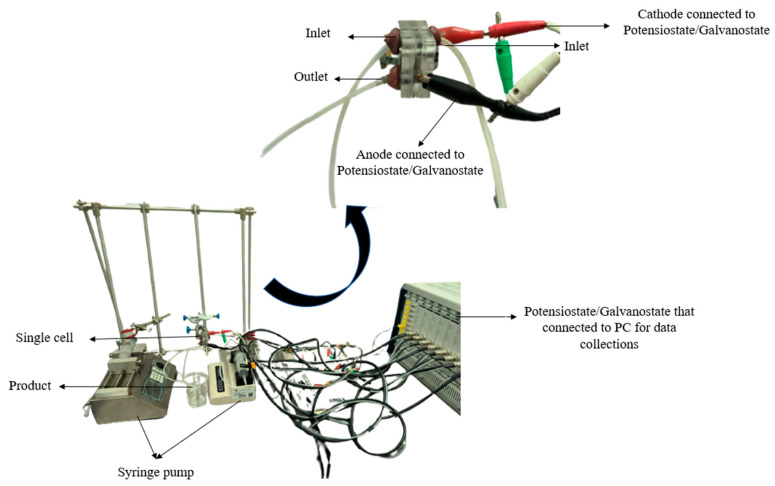
System setup of membrane-less DMFC for performance testing.

**Figure 3 micromachines-14-01247-f003:**
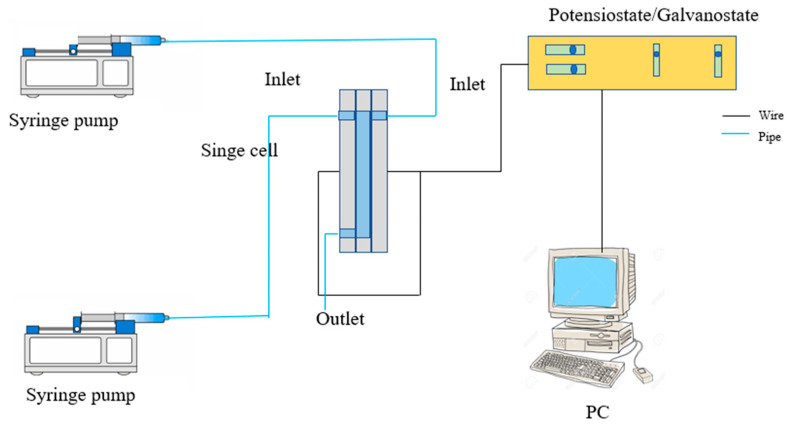
Schematic diagram for the membrane-less DMFC setup.

**Figure 4 micromachines-14-01247-f004:**
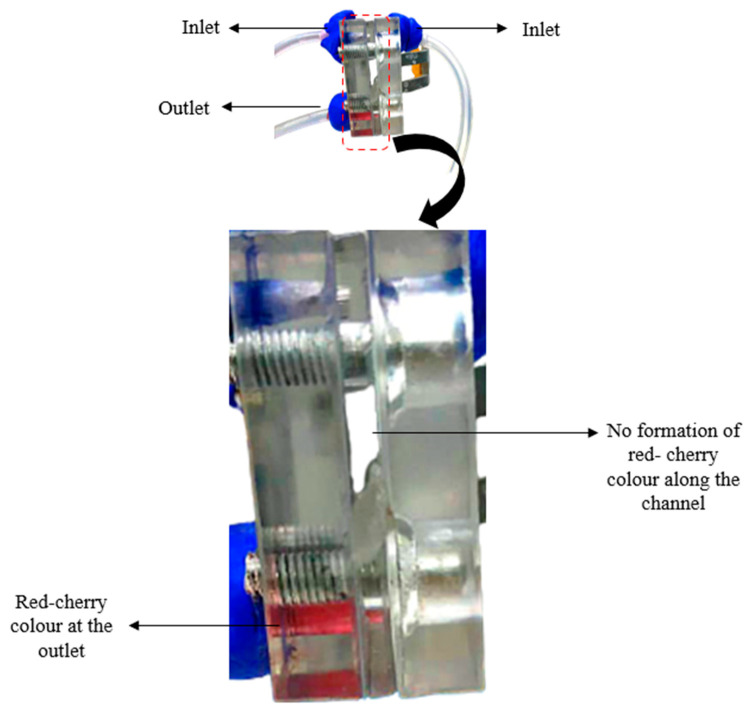
An observation of laminar flow shows no red-cherry color formed along the channel.

**Figure 5 micromachines-14-01247-f005:**
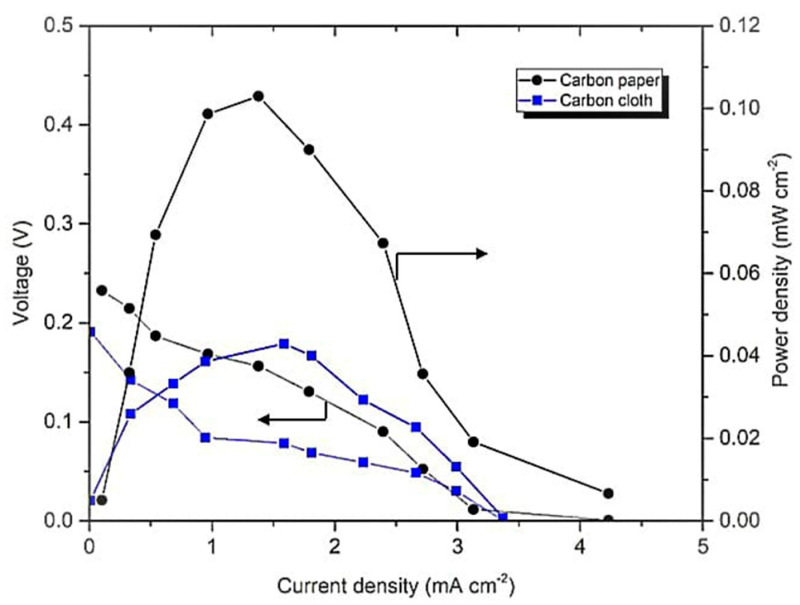
Polarization curve of F-shaped design of membrane-less DMFC-operated carbon cloth and paper.

**Figure 6 micromachines-14-01247-f006:**
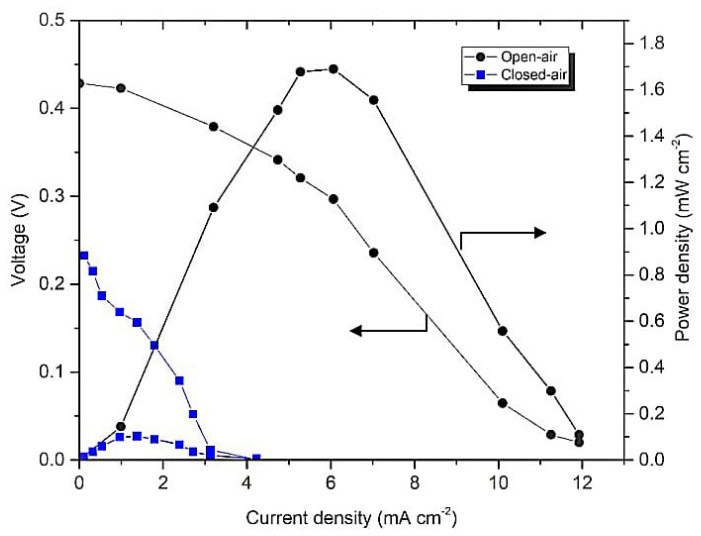
Polarization curve of F-shaped design of membrane-less DMFC operated with open- and closed-air at 1 M methanol and 0.5 M of sulfuric acid.

**Figure 7 micromachines-14-01247-f007:**
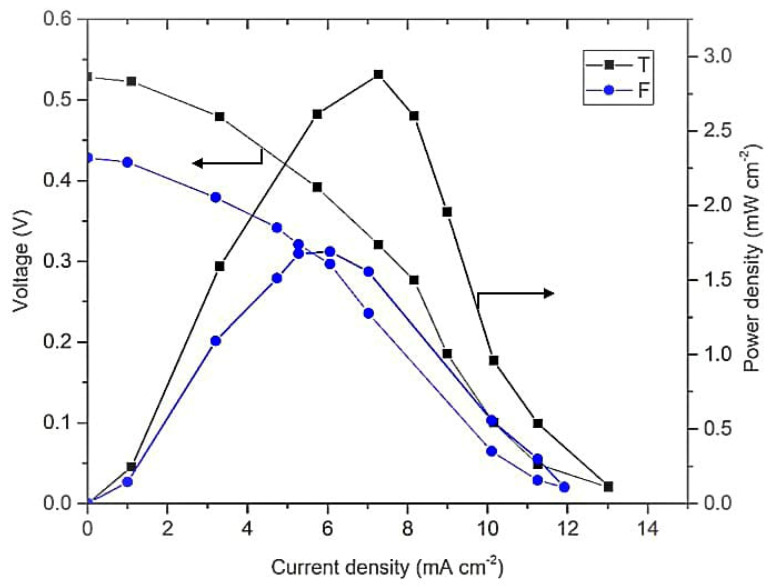
Polarization curve of F and T-shaped design of membrane-less DMFC operated at 1 M methanol and 0.5 M of sulfuric acid in open-air condition.

**Figure 8 micromachines-14-01247-f008:**
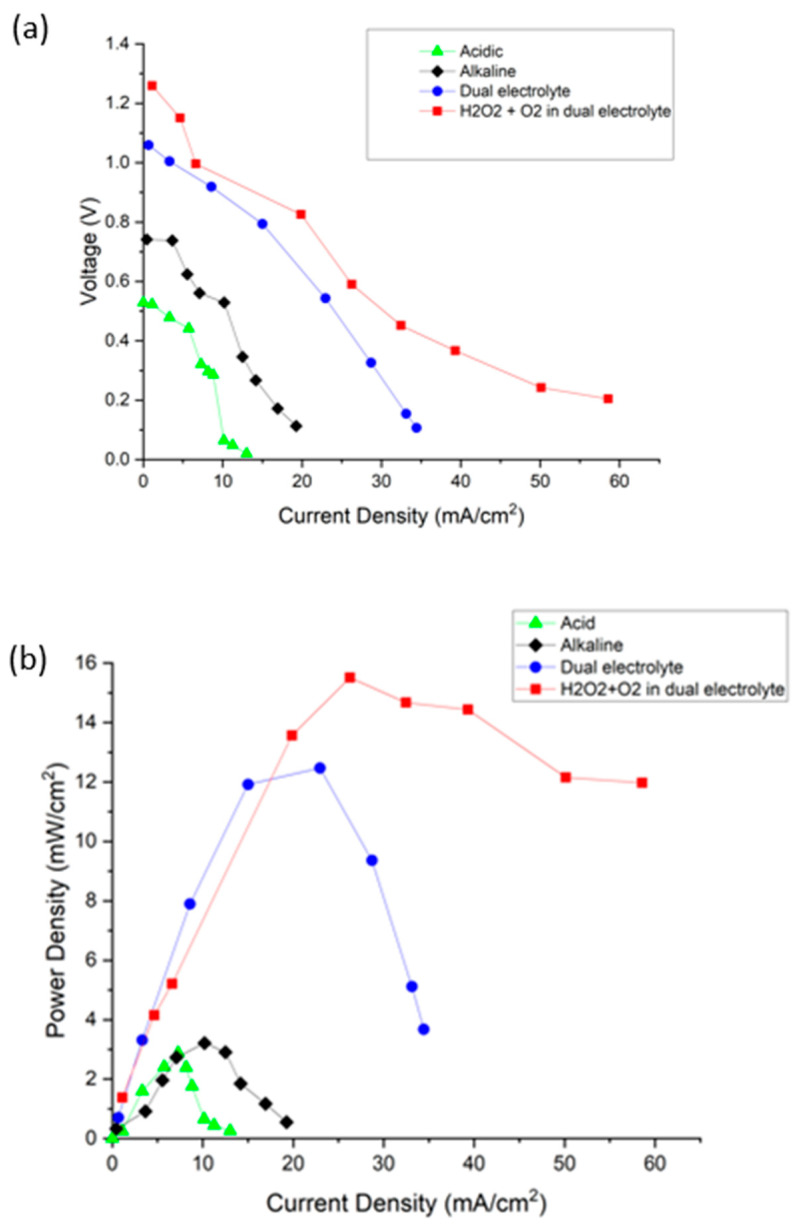
(**a**) Polarization and (**b**) power density curve of different environment for air-breathing membrane-less DMFCs at room temperature and pressure.

**Figure 9 micromachines-14-01247-f009:**
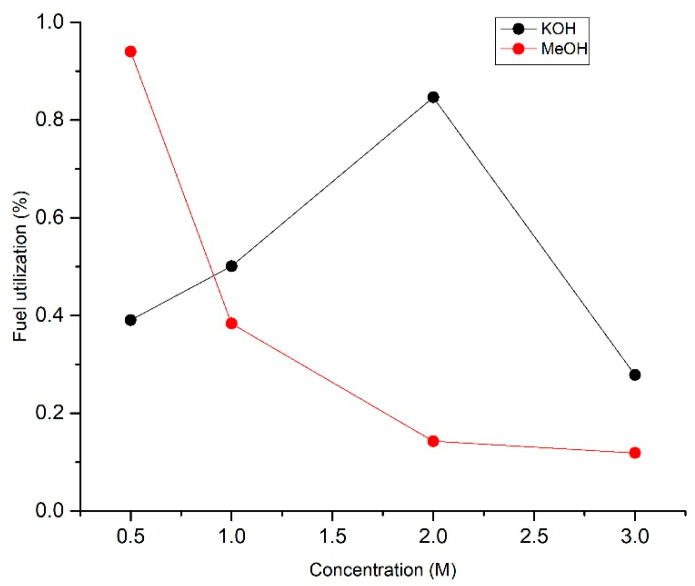
Fuel utilization of membrane-less DMFCs operated in mixed oxidants and dual electrolytes based on different concentration of KOH (
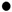
) and MeOH (
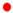
) in anode.

**Figure 10 micromachines-14-01247-f010:**
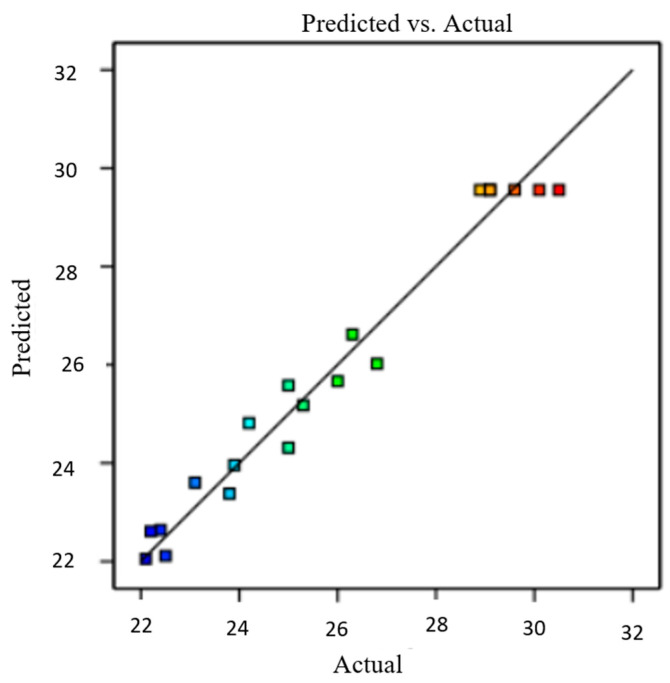
Comparison between experimental and predicted values for power density generation.

**Figure 11 micromachines-14-01247-f011:**
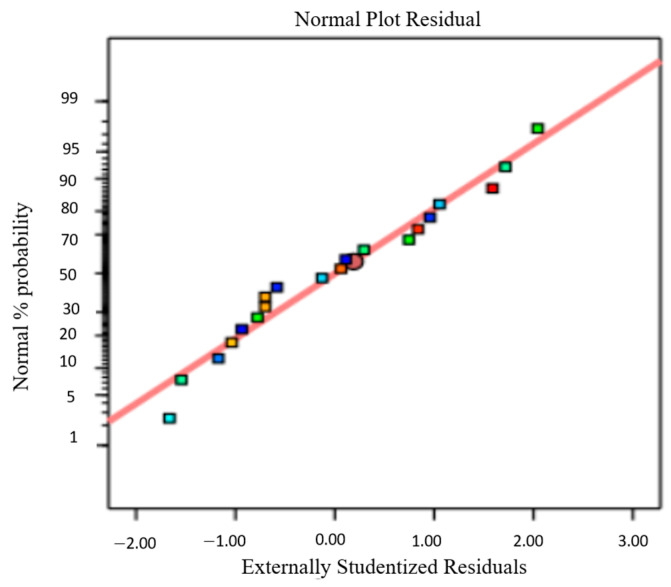
Normal probability plot of residual for power density generation.

**Figure 12 micromachines-14-01247-f012:**
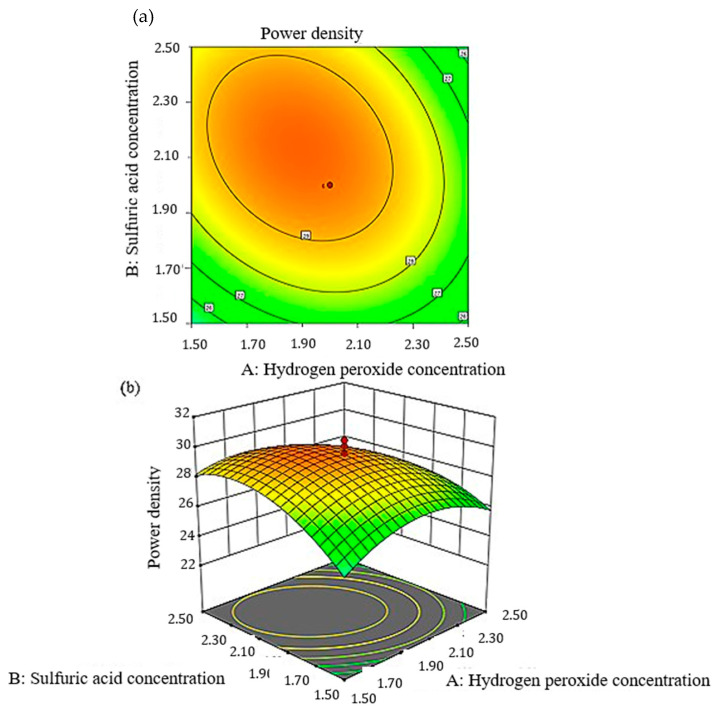
Plot for peak power density: (**a**) two-dimensional response surface, (**b**) three-dimensional response surface as a function of sulfuric acid and hydrogen peroxide concentration.

**Figure 13 micromachines-14-01247-f013:**
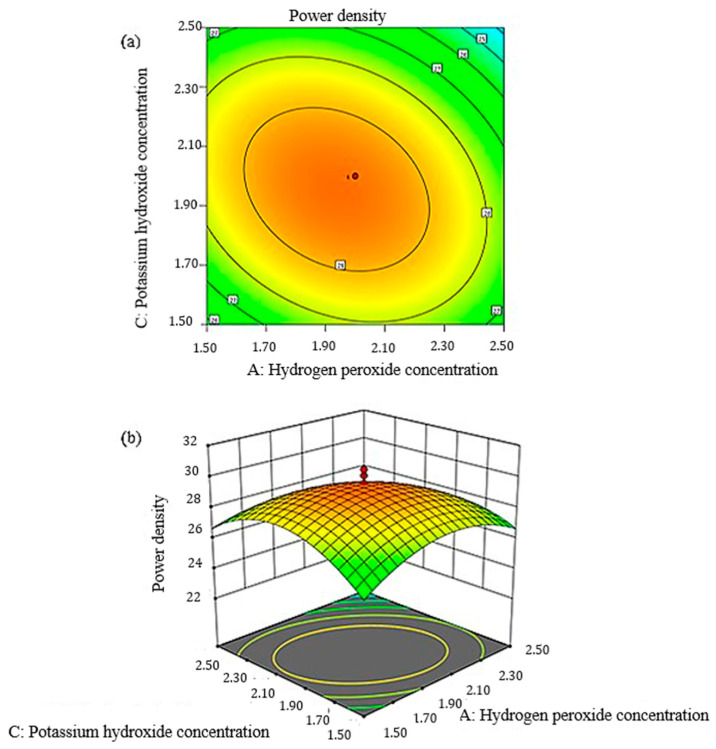
Plot for peak power density: (**a**) two-dimensional response surface, (**b**) three-dimensional response surface as a function of potassium hydroxide and hydrogen peroxide concentration.

**Figure 14 micromachines-14-01247-f014:**
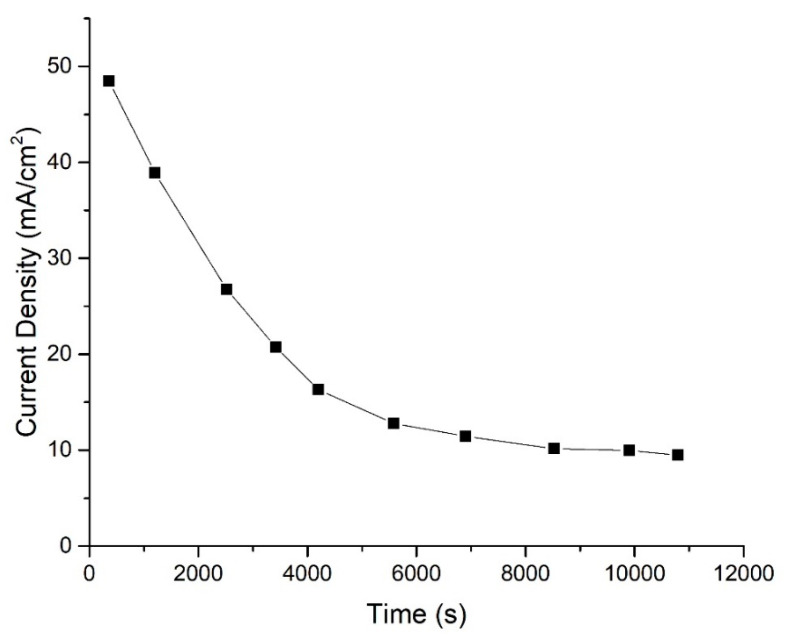
Chronoamperometric curve for membrane-less DMFC operation measured at 0.6 V cell voltage using the 1 M MeOH + 1.965 M KOH as anolyte and 2.130 M H_2_SO_4_ + 1.903 M H_2_O_2_ (parameter suggested by CCD).

**Table 1 micromachines-14-01247-t001:** Set up of membrane-less DMFC based in different environments.

Environment	Anolyte	Catholyte
All acidic	1 M MeOH + 0.5 M H_2_SO_4_	1 M H_2_SO_4_ + O_2_
All alkaline	1 M MeOH + 1 M KOH	1 M KOH + O_2_
Dual-electrolyte (Mixed-medium)	1 M MeOH + 1 M KOH	0.5 M H_2_SO_4_ + O_2_
Combination of O_2_ and H_2_O_2_ in dual-electrolyte	1 M MeOH + 1 M KOH	0.5 H_2_SO_4_ + O_2_ + 0.5 M H_2_O_2_

MeOH—Methanol. H_2_O_2_—Hydrogen peroxide. KOH—Potassium hydroxide. H_2_SO_4_—Sulfuric acid. O_2_—Oxygen.

**Table 2 micromachines-14-01247-t002:** Result of preliminary study using one-factor-at-a-one-time (OFAT).

Parameters	Varying Parameter (M)	Power Density (mW cm^−2^)
Varying: H_2_O_2_ concentration Fixed: H_2_SO_4_: 1 M, KOH: 1 M, MeOH: 1 M	0.5	12.7
	1	15.5
	2	17.9
	3	16.7
Varying: H_2_SO_4_ concentration Fixed: H_2_O_2_: 2 M, KOH: 1 M, MeOH: 1 M	0.5	17.0
	1	18.9
	2	21.4
	3	19.8
Varying: KOH concentration Fixed: H_2_O_2_: 2 M, H_2_SO_4_: 2 M, MeOH: 1 M	0.5	19.9
	1	21.4
	2	25.4
	3	23.9
Varying: MeOH concentration Fixed: H_2_O_2_: 2 M, H_2_SO_4_: 2 M, KOH: 2 M	0.5	24.8
	1	25.4
	2	27.1
	3	26.6

**Table 3 micromachines-14-01247-t003:** Factors for the response surface methodology.

Factor	Units	Low Level (−1)	High (+1)
A: H_2_O_2_ concentration	M	1.5	2.5
B: H_2_SO_4_ concentration	M	1.5	2.5
C: KOH concentration	M	1.5	2.5

**Table 4 micromachines-14-01247-t004:** Result of power density generation from CCD design.

Std. Order	A	B	C	Response, Y: Power Density
1	2.00	2.00	2.00	30.5
2	2.00	2.84	2.00	26.8
3	1.50	1.50	1.50	22.4
4	1.50	2.50	2.50	26.3
5	2.00	2.00	2.84	22.2
6	2.00	1.16	2.00	23.1
7	2.50	2.50	1.50	24.2
8	2.00	2.00	1.16	25
9	1.50	1.50	2.50	23.8
10	2.00	2.00	2.00	30.1
11	2.00	2.00	2.00	29.1
12	2.50	1.50	2.50	22.5
13	2.84	2.00	2.00	23.9
14	1.50	2.50	1.50	25
15	2.00	2.00	2.00	29.6
16	2.00	2.00	2.00	28.9
17	2.00	2.00	2.00	29.1
18	1.16	2.00	2.00	26
19	2.50	1.50	1.50	25.3
20	2.50	2.50	2.50	22.1

**Table 5 micromachines-14-01247-t005:** Analysis of variance (ANOVA) for quadratic model.

Source	Sum of Squares	df	Mean Square	F Value	*p*-Value Prob > F	
Model	150.92	1	16.77	34.46	<0.0001	significant
A-Hydrogen peroxide	3.52	1	3.52	7.23	0.02227	
B-Sulfuric acid	7.06	1	7.06	14.52	0.0034	
C-Potassium Hydroxide	3.50	1	3.50	7.18	0.0231	
AB	5.45	1	5.45	11.19	0.0074	
AC	7.22	1	7.22	14.84	0.0032	
BC	0.0450	1	0.0450	0.0925	0.7673	
A^2^	40.57	1	40.57	83.36	<0.0001	
B^2^	40.57	1	40.57	83.36	<0.0001	
C^2^	66.93	1	66.93	137.54	<0.0001	
Residual	4.87	10	0.4866			
Lack of fit	2.83	5	0.5662	1.39	0.3630	not significant
Pure Error	2.04	5	0.4070			
Correlation total	155.79	19				
Std. Dev.	0.6976		R^2^	0.9688		
Mean	25.80		Adjusted R^2^	0.9407		

**Table 6 micromachines-14-01247-t006:** Comparison between the experimental result and the predicted data.

H_2_O_2_	H_2_SO_4_	KOH	Power Density mW cm^−2^					Error %
			Prediction	Exp 1	Exp 2	Exp 3	Average	
1.903	2.130	1.965	29.717	28.7	29.2	29.5	29.72	1.98

**Table 7 micromachines-14-01247-t007:** The performance of membrane-less DMFC.

Medium	Fuel	Oxidant	Power Density (mW cm^−2^)	References
Dual-electrolyte	1 M MeOH	O_2_ and H_2_O_2_	29.72	This study
Alkaline	1 M MeOH	O_2_ and H_2_O_2_	3.8	[22]
Acidic	2 M MeOH	O_2_	7.4	[36]
Dual-electrolyte	1 M MeOH	O_2_	12.0	[17]

O_2_—Oxygen; MeOH—Methanol; H_2_O_2_– Hydrogen peroxide.

## Data Availability

Not applicable.

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
