# Peer review of "Optimization of Multiple Reactants in a Membrane-Less Direct Methanol Fuel Cell (DMFC)"

_micromachines, 2023, doi:10.3390/mi14061247_

Round 1

Reviewer 1 Report

In this manuscript, authors conducted a comprehensive optimization on multiple reactants in membrane-less DMFC. Overall, this is a good work, and I would like to recommend acceptance of this manuscript after several minor revisions:

1.      Page 3, Section 2.2, ‘15mm long and 3mm and wide window’ should be ‘15mm long and 3mm wide window’;

2.      Please explain that how the results in Table 2 are obtained?

3.      Page 14, the Eq. 11 should be rewritten;

4.      Fig. 6 (a), please give a detailed explanation on the obvious difference in the open circuit voltages (OCVs).

Reviewer 2 Report

The work presented to Micromachines deals with the utilization of a membraneless configuration for a methanol fuel cell. The paper is interesting, since it propose a novel configuration, but it has several flaws that must be corrected before the work could be accepted for publication. 

·      The templated employed reports “acoustic” in the upper part of each page, isn’t the journal micromachines?

·      A scheme representing the principle of the membraneless fuel cell when it is firstly described would be helpful for the reader. 

·      As well, it could be good a graphical comparison with the conventional PEM-FC (https://doi.org/10.3390/en15103588).

·      The figures employed in the first part of the manuscript are low quality. In Figure1, the copyright logos are visible under the images of the PC for the acquisition. In Figure2, the photos are really blurred and there is a shadow under the text.

·      As optical analysis for the determination of the absence of mixing within the channel of the fuel cell, it would have been better if a colorless fluid and a colored fluid were used. Furthermore, the mixing is not only related to the fluid dynamic of a system, whether to the affinity of the two fluids too, as well as the solubility of one fluid into the other. How can the authors be sure that the combo FeCl2 + PBS resembles the behavior of the anolyte and catholyte?

·      Figure3 is not readable: besides the differentiation of carbon paper and carbon cloth, which curve should be read on which axe? Same applies to Figure4 and 5.

·      Figure6 is completely blurred, and the legend is not readable

·      Conclusions should be rewritten. The section of optimization is not even discussed in the conclusions, despite the authors discuss how the results obtained with the present configuration are quite disappointing (section 3.6). 

Round 2

Reviewer 2 Report

I appreciated the authors responses, and I believe the work can now be published in micromachines